# Polysaccharides Composite Materials as Carbon Nanoparticles Carrier

**DOI:** 10.3390/polym14050948

**Published:** 2022-02-26

**Authors:** Magdalena Krystyjan, Gohar Khachatryan, Karen Khachatryan, Marcel Krzan, Wojciech Ciesielski, Sandra Żarska, Joanna Szczepankowska

**Affiliations:** 1Faculty of Food Technology, University of Agriculture in Krakow, Al. Mickiewicza 21, 31-120 Kraków, Poland; karen.khachatryan@urk.edu.pl; 2Jerzy Haber Institute of Catalysis and Surface Chemistry, Polish Academy of Sciences, 30-239 Krakow, Poland; marcel.krzan@ikifp.edu.pl; 3Institute of Chemistry, Jan Dlugosz University in Czestochowa, 13/15 Armii Krajowej Ave., 42-200 Czestochowa, Poland; w.ciesielski@interia.pl (W.C.); zdanowskasandra@gmail.com (S.Ż.); 4Faculty of Biotechnology and Horticulture, University of Agriculture in Krakow, Al. Mickiewicza 21, 31-120 Krakow, Poland; j.szczepankowska17@wp.pl

**Keywords:** carbon, graphene, carbon nanotubes, carbon quantum dots, nanoparticles, nanostructures, polysaccharides

## Abstract

Nanotechnology is a dynamically developing field of science, due to the unique physical, chemical and biological properties of nanomaterials. Innovative structures using nanotechnology have found application in diverse fields: in agricultural and food industries, where they improve the quality and safety of food; in medical and biological sciences; cosmetology; and many other areas of our lives. In this article, a particular attention is focused on carbon nanomaterials, especially graphene, as well as carbon nanotubes and carbon quantum dots that have been successfully used in biotechnology, biomedicine and broadly defined environmental applications. Some properties of carbon nanomaterials prevent their direct use. One example is the difficulty in synthesizing graphene-based materials resulting from the tendency of graphene to aggregate. This results in a limitation of their use in certain fields. Therefore, in order to achieve a wider use and better availability of nanoparticles, they are introduced into matrices, most often polysaccharides with a high hydrophilicity. Such composites can compete with synthetic polymers. For this purpose, the carbon-based nanoparticles in polysaccharides matrices were characterized. The paper presents the progress of ground-breaking research in the field of designing innovative carbon-based nanomaterials, and applications of nanotechnology in diverse fields that are currently being developed is of high interest and shows great innovative potential.

## 1. Introduction

One of the most promising and dynamically developing fields of science is nanotechnology. It can be defined as a set of techniques that control objects at the nanometer scale to produce materials and devices with unprecedented properties [1]. Among the forerunners of nanotechnology was American physicist Richard Feynman. In his famous lecture entitled “There’s plenty of room at the bottom”, he introduced the idea of miniaturization and presented new possibilities inherent in techniques that can manipulate matter at the level of atoms and molecules. However, it is worth noting that the concept of nanotechnology appeared much earlier, which laid the foundation for the field of science that is nanotechnology [2,3,4].

Nanometric structures are assumed to have at least one dimension, ranging from 1 to 100 nm. However, these are only contractual values. In practice, we start talking about nanomaterials when we cross a threshold below which differences in physical, chemical and biological properties are observed in comparison to materials of the same chemical composition but occur on a macrometric scale [1,3,4,5]. The unique characteristics of nanomaterials are due to the fact that below certain sizes, quantum effects begin to significantly affect the behavior of a particle. Factors such as the size and shape of individual structural elements may be more indicative of the specifics of a particle than its chemical composition.

The larger surface-to-volume ratio and large specific surface area in nanometric materials mean that their properties are determined mainly by atoms and interactions located on their outer layer, rather than deep within the phase, as is the case for macroscopic objects. In addition, the number of chemical bonds is reduced, so that individual bonds gain a greater influence on the functionalities of the nanostructure [4,6]. Depending on the number of dimensions, where their magnitude is expressed in nanometers, zero-dimensional (0-D), one-dimensional (1-D), two-dimensional (2-D), and three-dimensional (3-D) materials can be distinguished. Zero-dimensional structures are spherical nanoparticles in which all three dimensions are on the order of a few to tens of nanometers. These types of nanostructures include colloidal nanoparticles, quantum dots and fullerenes. In one-dimensional systems, the quantum confinement effect occurs only in two mutually perpendicular directions. Carbon nanotubes, graphene nanowires, metal nanowires or polymer nanofibers, among others, are characterized by such spatial organization. When only one of the three dimensions of a structure is restricted, we are dealing with a two-dimensional form. This includes various types of nanoplates and nanolayers, as well as monomolecular (monoatomic) layers, which have a thickness corresponding to the diameter of one atom. In turn, 3-D nanostructures are homo- and heterogeneous materials built from monoblocks with nanometer-scale sizes that form common interfaces. Photonic crystals are an example of 3-D structures [7,8,9].

The properties of carbon nanoparticles do not always allow for their direct application. This is due to the variety of materials, their different properties, which in many cases leads to a lack of compatibility with carbon particles. One of the difficulties in the synthesis of graphene-based materials, which have limited their use in many fields, is their tendency to aggregate due to the presence of Van der Waals interactions between graphene layers [10,11]. Therefore, in order to achieve a wider use and better availability of nanoparticles, they are introduced into matrices, most often polysaccharides with high hydrophilicity. Due to the diversity of functional groups, polysaccharides are an attractive carrier for carbon nanoparticles. The advantages of polysaccharides are their low cost, availability, ease of acquisition and processing. An additional advantage is their biodegradability and renewability. Importantly, polysaccharides provide a green alternative to synthetic polymers in the processing of nanomaterials [12].

The incorporation of a nano-additive into a polysaccharide matrix is intended to improve the mechanical, thermal, optical, surface or biological properties of the composite. They have found wide application in various fields: medicine, pharmacology, tissue engineering, optoelectronics, microelectronics, as well as in biological systems and as antimicrobial agents. An important aspect is that these materials can be obtained in an environmentally friendly way, without the addition of toxic reagents. Numerous studies have proven the bioactivity of polysaccharides and enabled their use in clinical practice, nutrition and dietetics [13,14]. Polysaccharides are characterized by a variety of properties, such as low, medium or high molecular weights, variable polydispersity, the formation of linear or branched structures, monofunctionality (compounds containing only hydroxyl groups) or polyfunctionality (compounds with hydroxyl, carboxyl and/or amino groups), a high degree of chirality, different water solubility, low toxicity and immunogenicity. It is also worth noting that some polysaccharides may exhibit antioxidant, immunomodulatory, anti-inflammatory, antiviral, antimutagenic, anticancer and anticoagulant activities [15,16,17,18]. As a result of these characteristics, polysaccharides have found wide applications in nanotechnology. Polysaccharides such as: cellulose, starch, chitosan, and alginates, are a focus of attention [19].

Polysaccharides can be found in many natural sources such as marine algae (e.g., alginate), plants (e.g., pectin, guar gum), bacteria (e.g., dextran, xanthan gum), and animals (e.g., chitosan, chondroitin). Most natural polysaccharides (e.g., chitosan, starch, alginate) contain hydrophilic groups (hydroxyl, carboxyl, amino), which show bioadhesion to biological structures (epithelia, mucous membranes). Nanoparticles formed from bioadhesive polysaccharides may prove useful as carriers of drugs, extending their persistence in a particular environment [19,20]. Considering that the type of matrix affects the properties and reactivity of nanocomposites, the polysaccharides most commonly used to produce bionanocomposites can be classified according to their ionic nature: neutral (starch, agarose, pullulan, dextran), anionic (carrageenans, hyaluronic acid, heparin, furcellaran) and cationic (quaternary chitosan).

## 2. Carbon-Based Nanostructures

The production of new varieties of carbon-based nanomaterials, the study of their properties, and their subsequent functionalization are important tasks of nanotechnology and materials engineering. Carbon nanoparticles are most commonly used as nanofillers in composite materials, as this combination can significantly increase their functionality. Their key functional properties include extremely good electrical and thermal conductivity, electrochemical and thermal stability, and mechanical strength. Recent studies have also shown their high biocompatibility with cells such as osteocytes and neurons and their antimicrobial activity, which greatly enhances their applicability in biomedical sciences [7,9,21,22].

There are several allotropic varieties of carbon. The well-known varieties are nanotubes, fullerenes, graphene and its derivatives. Figure 1 shows the carbon structures: (A) nanotubes, (B) graphene and (C) quantum dots. Due to differences in the spatial arrangement of the elements, each variant is characterized by different physical properties and different chemical activities. Graphene is a single layer of graphite, one atom thick, in which the carbon molecules form a hexagonal arrangement. It is a structural element of other carbon materials, including nanotubes and fullerenes. Nanotubes are cylindrical molecules composed of rolled sheets of graphene. We can distinguish between single-walled nanotubes, consisting of a single layer of graphene, and multi-walled nanotubes, made up of several concentrically connected nanotubes. Fullerenes, on the other hand, are molecules with a spherical or ellipsoidal structure, composed of an even number of carbon atoms. They are the only pure allotropic variety, containing no admixtures of other elements. The valence orbitals of all three forms are in the sp^2^ hybridization state. This means that each carbon atom forms three bonds in one plane, spaced at 120 degrees. In turn, the fourth electron is outside the plane of the orbital, freely moving as an electron gas. Consequently, this has implications for the high strength of carbon nanostructures and their conductive properties [1,3,6,7,23,24,25,26].

### 2.1. Graphene

Graphene is the strongest material discovered so far. The strength of a defect-free graphene layer is up to 130 GPa. In contrast, the value of Young’s modulus, which determines elasticity in tension and compression, is about 1 TPa. Graphene is thus a much stronger structure than steel of the same thickness and up to a thousand times stronger than diamond. In addition, it has the highest thermal and electrical conductivity [26,27,28,29]. Recent studies have also shown its germicidal and bacteriostatic properties, almost complete impermeability to gases, and ability to absorb visible and near-infrared light [11,30]. However, graphene itself has some limitations. These include a high tendency to agglomerate, hydrophobicity and low reactivity. In order to improve its properties, the surface is functionalized by modifying the bonds and bonding various chemical groups [10,11]. Graphene oxide is the most commonly used graphene derivative in science and industry. In its structure, it contains randomly distributed hydroxyl and epoxy groups, as well as carboxyl and carbonyl groups at the edges of the layer. Due to the oxygen-functional groups incorporated, it acquires hydrophilic properties and can easily disperse in water and other polar substances. In addition, bonds formed between carbon and oxygen atoms acquire sp^3^ hybridizations. This leads to an increase in the number of interactions that can occur on its surface, but also makes its electrical conductivity decrease [10,28,31]. Due to the presence of surface polar groups, GO interacts with biomolecules such as proteins and lipids. This affects its interesting properties: a good diffraction strength, high thermal and electrical conductivity, mobility of charge carriers and biocompatibility, light weight, high Young’s modulus, large planar surface, and electron delocalization [11,32,33].

Graphene oxide can be reduced chemically, thermally or electrochemically [26]. In this way, so-called reduced graphene oxide is obtained. It differs from graphene in that it contains functional group residues and numerous defects in the crystal lattice. As a result, it has weaker mechanical and conductive properties that are, however, still better than graphene oxide [31]. Due to its much simpler production process compared to graphene, it is a good compromise between graphene oxide and graphene itself and is a form often used in the industry [27].

### 2.2. Carbon Nanotubes

Rolling graphene sheets into a cylindrical tube form partially changes its properties. Carbon nanotubes, which do not contain a significant number of structural defects, exhibit strengths close to 100 GPa, and their stiffness can reach 1000 GPa. Some studies have shown an increase in Young’s modulus of more than 1000% after the use of carbon nanotubes [34]. Their tensile strength is, therefore, up to 400 times higher than that of steel. They also have extremely high thermal conductivity, ranging from 2000 to 6000 W m^−1^ K^−1^ [35]. In addition, they have a very low density of approx. 1.3 g·cm^3^ and high resistance to high temperatures (up to 2800 °C in vacuum). Nanotubes can withstand an electric current density several times greater than that of copper: around 4 × 10^9^ A cm^−2^ [36]. However, these parameters are not constant and can depend on many factors. One of them is chirality, which determines the optical properties of nanotubes. Compounds exhibiting chirality are optically active, that is, they can rotate the plane of polarized light by a certain angle. The significant parameter that defines the geometrical structure of carbon nanotubes is the chiral angle. This angle classifys the carbon nanotubes into three geometrical shapes: armchair (θ = 30°), chiral (0 < θ < 30°), and zigzag (=0°). Only the last form is optically active. The other two configurations are achiral. One third of the chiral nanotubes have metallic properties, the rest are semiconductors [37,38].

There will be differences in properties depending on whether it is a single-walled (SWCNT) or multi-walled (MWCNT) form. Nanotubes composed of multiple layers have more structural defects, resulting in poorer mechanical properties. They exhibit lower stiffness and flexibility [34,39]. The conductive properties of single-walled and multi-walled nanotubes are also different. The ability to conduct electricity in SWCNTs can vary depending on the degree of chirality. In the case of MWCNTs, they always have at least as high a conductivity level as metals.

### 2.3. Carbon Quantum Dots (CQDs)

Carbon quantum dots (CQDs) are nanomaterials around 10 nm in size, which have gained great popularity in recent years and are the subject of many studies. Due to the diversity of both physical and chemical synthesis methods, chemical inertness, low toxicity, photo-induced electron transfer, highly tunable photoluminescence behavior and biocompatibility make carbon dots a promising replacement for semiconductor quantum dots. CQDs find applications in industries such as LEDs [40,41,42], biosensors [43,44] and supercapacitors [45,46]. They possess highly desirable properties as semiconductor nanoparticles and are promising nanomaterials for photocatalysis, ion sensing, biological imaging, heavy metal detection, adsorption treatment, supercapacitors, membrane fabrication and water purification. Due to their small size that allows them to penetrate cell membranes, they have great potential for biomedical applications, especially as drug transporters. Sustainable raw materials are widely used to produce CQDs because they are cost-effective, environmentally friendly, and effectively minimize waste production. CQDs can be produced by laser ablation, microwave field, hydrothermal reaction, electrochemical oxidation, the reflux method and by ultrasound. These methods are subject to a combination of several chemical reactions such as oxidation, carbonization, pyrolysis and polymerization [47,48].

In fact, most organic compounds rich in carbon atoms can be used as raw material for the synthesis of CQDs [49]. According to previous studies, CQDs have been produced from various natural carbon sources such as citric acid, denatured milk, dried leaves, broccoli, food waste, pomelo fruit, ginkgo leaf, grass, humic acid, ascorbic acid, gelatin and straw [44,50,51]. The aforementioned pathways to obtain CQDs are inexpensive, fast and scalable [52]. A study was carried out to synthesize CQDs with different functional properties, controlling the obtained size of the carbon dots and the chemical activity by modifying the functional groups on the surface [53], which depend on the concentration and type of precursors used.

CQDs are usually classified as zero-dimensional substances that exhibit specific optical properties. They typically exhibit optical absorption in the UV region with a tail extending into the visible range. One of the most fascinating features of CQDs, both from a fundamental and application-oriented perspective, is their photoluminescence (PL). The study revealed that one unique feature of photoluminescence of carbon quantum dots is the wavelength dependence and emission intensity on λ_ex_. This is likely due to quantum effects or different emission traps resulting from the structural and surface functional diversity of CQDs [53]. The photoluminescence region of CQDs is very wide, starting from the ultraviolet and moving to the near infrared. Although the luminescence behavior of carbon nanodroplets is currently still not fully clear, two factors are believed to play a key role, namely the quantum size below 10 nm and surface defects that can act as excitation energy traps.

## 3. Application of Carbon Nanoparticles

The last few years have seen an explosion in the use of nanotechnology in many areas of science and industry: from electronics to biotechnology. One of the most numerous groups of raw materials currently used in nanotechnology are carbon materials. The discovery of carbon allotropes, including fullerene, carbon nanotubes, graphene, and carbon quantum dots, has revolutionized industry. Since the emergence of nanotechnology, nanocomposites based on carbon allotropes have become a leading sector of research and progress due to their unique mechanical, optical and electrical properties.

Medicine is an area where nanotechnology seems to have an exceptionally large impact. Nanodevices and nanomaterials have great potential in the diagnosis, monitoring and treatment of diseases. This is primarily due to the ability of nanostructures to penetrate into the human cells, thus enabling targeted therapy and the accurate detection of the disease source. The precision of nanotechnology makes it a potential technology for cancer detection and therapy. In the diagnosis of these diseases, nanoparticles are used as contrast agents to detect cancer cells at an early stage and monitor the progress of treatment. This can be carried out with fluorescent tracers, which emit light in the visible range when exposed to UV light of the appropriate wavelength, or with magnetic tracers, where detection is performed using magnetic resonance imaging. There is much research using carbon nanomaterials as biosensors to monitor the concentration of marker substances in the blood. Particles used for cancer detection can simultaneously be used to treat cancer. The currently used therapies, due to their low specificity, also display a high cytotoxicity against healthy cells. Nanotechnology offers the opportunity to bypass this problem, thereby reducing the side effects of therapy. Placed in nanoparticles, chemotherapeutics can be delivered to cancer cells with remarkable precision. They also allow higher doses of the drug to be administered, which are still safe doses for the patient [54,55]. The research on the use of nanoparticles in the fight against cancer is current and in a dynamic phase. Many nanopreparations are already in the final stages of clinical trials. This gives great hope that much safer and more effective methods of fighting these diseases will be introduced in the coming years.

Another important area of nanotechnology applications in medicine is tissue engineering. Until now, it has mainly been used to fabricate chemically inert scaffolds for damaged tissues or whole organs. However, recent studies have shown that the use of biologically active materials can significantly improve the regeneration of damaged organs. This is made possible by the interaction between cells and tissue replacements and by the release of factors that promote repair. Carbon nanostructures such as carbon nanotubes and graphene can help with functionalization and create more compatible scaffolds. Nanotubes provide good electrical conductivity, increased elasticity and maximum loading, which is particularly useful in cardiac tissue engineering [44,56]. Graphene also has great potential in bone regenerative medicine. It exhibits very good mechanical properties and a high adhesion. Moreover, the presence of graphene oxide in biopolymer-based scaffolds stimulates bone tissue mineralization and cell growth and proliferation on the scaffold surface [57].

CQDs, in turn, are widely used in drug delivery, gene delivery, biodetection and even bioimaging [58,59]. The bactericidal and bacteriostatic effect of materials containing carbon dots has also been proven [60]. Their potential has also been verified in multifunctional diagnostic platforms, cellular and bacterial bioimaging, the development of theragnostic nanomedicine, and many other applications [61].

Nanotechnology also has a number of applications in the food industry. Nanoparticles can be used as food additives to improve the texture, taste, smell and consistency, and increase the nutritional value of products. Another area where it can provide many innovative solutions is the production of modern, functional food packaging. Currently, synthetic polymers, which are not biodegradable, are most often used, and thus pose a serious problem for the environment. An alternative to traditional materials is to create composites made of naturally derived polymers that are biodegradable, biocompatible and completely safe for humans. This type of packaging could be produced using easy, cost-effective and eco-friendly techniques. As the food industry generates a huge amount of waste, these solutions would significantly help to reduce environmental pollution [62,63,64,65]. Biopolymers used as matrices in nanoporous films should be inexpensive, nontoxic, readily available, and completely degradable. For these reasons, polysaccharides, in particular chitosan, cellulose, alginate, starch and pectin, are of major interest [64]. Carbon nanomaterials are often used as nanofillers. They give the composites suitable mechanical and optical properties as well as gas barrier properties [63]. Polymer composites, when enriched with nanofillers, acquire a number of new properties that allow them to be used as so-called “active”, “smart” and “enhanced”. Such “improved” composites exhibit antimicrobial and antioxidant activities, prevent harmful enzymatic transformations, and protect food from UV radiation. They can be used as nanosensors to detect microbiological and biochemical changes in food products when used in packaging [63,64,65,66,67]. Integrating biological analytical receptors with different types of nanomaterials can significantly increase their sensitivity and specificity. Such indicators can provide immediate visual and qualitative information about the condition of food by changing the color or altering its intensity. The high sensitivity of nanosensors to toxins, viruses, pesticides and pathogens has been confirmed by numerous studies in recent years [10,11,68,69]. They are increasingly used in food and water analysis, replacing the time-consuming, costly and invasive analytical techniques currently used to monitor product suitability. In addition, nanosensors can provide information on environmental changes such as temperature and humidity, providing a complete view of the condition of the food [63,66,70].

Carbon dots can be used as additives in active packaging materials to improve physicochemical properties such as mechanical, barrier, antioxidant, antimicrobial and light-blocking properties. They can also be used as sensors in packaging materials to provide quality or food safety information [71]. CQDs play important roles in active food packaging materials. For example, they can be used to improve the mechanical, barrier or optical properties of films [72,73,74]; protect packaged food through their antioxidant and antimicrobial properties [75,76]; prevent counterfeiting by creating unique labels [77]; and detect changes in food quality during storage [78].

A broad area of research interest is the use of carbon nanoparticles in energy storage. Carbon nanotubes (CNT) and their derivatives have great potential in electrical and electronic applications such as photovoltaics, sensors, semiconductor devices, displays, conductors, smart materials and energy conversion devices (for example fuel cells, batteries and supercapacitors) [79,80].

While the plastics and composites industry accounted for the largest share of the overall use of carbon nanotubes in 2021, the energy storage market, driven by demand for electrification, is growing rapidly [81]. Nanotubes are a key conductive additive for the anode and cathode in both current and new generation lithium-ion batteries, and the inclusion of a relatively small amount of CNTs can significantly increase the energy density of these batteries. The way in which the CNTs are dispersed, their use with or without a binder, and their combination with other materials undoubtedly affect their electrochemical performance. The developments in the energy market of natural resources (i.e., crude oil, natural gas, etc.) and forecasts for energy needs make it necessary to seek new alternative sources of energy and materials for the storage of CNTs [82]. One of the most popular methods of energy storage is lithium-ion batteries (Li-ion), commonly available in portable electronic devices such as smartphones, laptops and power banks. Li-ion batteries have seen only a 1.6-fold increase in capacity in the last 24 years (according to the IDTechEx study: “Flexible, Printed and Thin Film Batteries 2015–2025”) [83]. The development of lithium-ion battery technology, leading to the extension of battery life, is slower than the development of the devices in which these batteries power them. As a result, there is a need to look for new cells that will store energy in a more efficient way, and at the same time will not be more expensive than the currently used ones and will be characterized by better electrochemical parameters. Batteries manufactured using commonly known technologies pose a risk of explosion—an explosion of a cell phone battery on board a Boeing [84] led to a ban on carrying some Samsung smartphones on board airplanes, as well as carrying them in checked baggage [85]. Electrode materials and devices based on carbon nanotubes can have a high power density, excellent specific capacity and are light, miniature, flexible and safe to use. CNTs used in the anode material, instead of traditional materials such as graphite, improve electrochemical performance by increasing the surface area and high electrical conductivity [86]. It was found that the functionalization of CNT surfaces, e.g., by introducing heteroatoms into the structure of multi-wall carbon nanotubes, improves their conductivity, stability and optimization of work in electrochemical devices, for everyday and specialist use [87,88,89]. Additionally, the modification of MWCNT with derivatives of tio- or selenophosphorous acids can form the basis for the construction of new electrochemical cells (as a basic component of electrodes). Such cells can potentially be characterized by a high efficiency, low resistance, low toxicity and small size, which may be useful in modernizing nanoelectronics and increasing the stability of the cell’s operation [90].

Currently a very promising solution is to store hydrogen and energy using the cells composed of ternary systems (alloys, composites, etc.). This storage of energy has the greatest prospect among the many methods proposed so far. They are developed lightweight alloys and have an intermetallic compounds layout of lithium Li-Me-O, which can be used as electrodes in batteries [91,92]. Light alloys (hybrid materials) and lithium intermetallic compounds of the Li-Mg-Si system, developed in recent years, are able to absorb a maximum of 8.8% by weight of hydrogen [93].

At present, this is the maximum value on a global scale (without taking into account hydrides such as: LiH, NaH, LiBH4, NaBH4 and others that are not based on intermetallic compounds). New hybrid materials made of intermetallic alloys/mono and multiwall carbon nanotubes (SWCNT and MWCNT) show better absorption properties due to the presence of extended surfaces of carbon components. So far, multicomponent composite alloys containing lithium and d-electron elements as materials for energy storage in batteries have been insignificantly studied worldwide. Previous studies demonstrated extremely beneficial effects of d-electron elements on thermal resistance and catalytic properties of the resulting materials and their electrochemical properties. To obtain hybrid materials (metal alloys are combined with CNT); magnesium alloys were used, which are the basis for the creation of light and ultra-light alloys. They are also used as materials for storing hydrogen, as well as materials for the construction of electrodes for metal hydride and magnesium-ion batteries. The search for new metal hydrides covers magnesium alloys with rare earth transition metals and doped with their p or s electron elements [94,95].

The synthesis and characterization of a new quaternary carbide, namely dimagnesium lithium aluminum carbide, Mg1.52Li0.24Al0.24C0.86, belonging to the family of hexagonal close-packed (hcp) structures was obtained. The presence of carbon improves the corrosion resistance of the Mg1.52Li0.24Al0.24C0.86 alloy compared to the ternary Mg1.52Li0.24Al0.24 alloy and Mg. The increase in corrosion resistance is due to the sealing of the structure and the strengthening of the bonds between atoms, which causes the transition from the centrosymmetric space group P63/mmc to the non-concentric space group P6m2 [96].

Lithium and magnesium alloys and their combination with rare earth transition metals and electron p or s and CNTs have great potential as energy storage materials, including for hydrogen storage or for building new generations of batteries. Research has shown that modifications of metal phases with carbon nanotubes also have a very positive effect. These modifications increase their specific capacity as well as their stability and resistance to external factors [97].

New energy storage materials based on lithium, magnesium or magnesium systems with rare earth metals or multi-wall carbon nanotubes as safe energy stores will contribute to the development of alternative energy sources for transport. These types of materials can also improve energy security and independence. At the same time, they will allow us to eliminate the basic problems that constitute a barrier to the widespread use of technologies, i.e., storage, transport and production.

For many years, research has been conducted on the interaction of MWCNT with polysaccharides that can be used, for example, as electrolytes in new types of batteries. One of the methods for obtaining these types of systems is by placing powders of CNT in aqueous solutions of various polysaccharides such as sacran, xanthan gum, and alginates and were dispersed under sonication. The obtained result could suggest the MWCNT-dispersing in sacran solution lowered the variety of wall numbers of MWCNT. These results show the fabrication of MWCNT-containing materials, which might show an absorbency, conductivity and many other functionalities [98]. This particularly involves their use as electrode binders, separators and gel/solid polymer electrolytes [99]. Additionally, MWCNT/starch composites were successfully prepared. It has been shown that they possess higher electrical conductivity, improved thermal stability and UV absorption compared to pure starch. The controlled synthesis of complexes of MWCNTs with starch is expected to pave the way for a much broader range of applications of the composites. The prepared MWCNT/starch composite could be used as electroactive polymer, biosensors, electronic device, orthopedic applications, and artificial arms in robotics, UV shielding, gas and flame protection and alternative of petroleum-based packaging. [100]. Therefore, in order to prepare research in this field, it is necessary to thoroughly understand the behavior of MWCNT as carriers and materials for energy storage; therefore, their properties in this respect are presented below. This will help in further research into MWCNT-polysaccharide composites.

## 4. Carbon Nanoparticles/Nanostructures in Polysaccharides Matrices

Nanotechnology applied to biological and biomedical challenges were the subject of extensive research in recent years. Functional nanomaterials that can transport biologically relevant molecules have become very useful in many scientific and industrial fields. Additionally, studies on biological materials that can serve as nanoparticles are a very timely topic. Among the natural polymers used to form nanocomposites, polysaccharides have attracted particular attention. They are readily available, biodegradable, safe for human use, film-forming, foam-forming, have good hydrogel-forming abilities, and provide a tight barrier to oxygen and carbon dioxide. Individual polysaccharides also exhibit many other desirable attributes, such as antibacterial or antioxidant properties. This makes the materials produced on their basis have a very wide range of applications. Depending on their origin, polysaccharides have various properties—antioxidant, immunomodulatory, anti-inflammatory, antiviral (including HIV), antimutagenic, anticancer and anticoagulant [101,102,103].

The study confirmed the usefulness of various polysaccharides in the synthesis of inorganic nanoparticles. Polysaccharides act as reducers and stabilizers—matrices that guarantee the formation of nanoparticles of uniform size, and thus fulfil the requirements of their practical applications. Nanoparticles immobilized in such matrices exhibit all the required properties: functionality, barrier properties, transparency and other interesting features. Nanocomposites with such nanoparticles are biodegradable and environmentally friendly. Thus, they have many potential applications including diagnostics, therapeutics and agricultural production [104]. The most commonly used polysaccharides in nanotechnology are starch, cellulose, alginates, pectins, xanthan gum, cyclodextrins, chitosan, heparin, furcellaran and hyaluronic acid [65,105]. Many studies have been carried out to improve the mechanical and barrier properties of polysaccharide-based films. For this, one of the possible ways is the modification of polymers with inorganic compounds. The enrichment of polysaccharide matrices with graphene oxide [11], nanotubes [106], and carbon dots [107] has been described.

### 4.1. Hydrogels

Due to their broad properties, hydrogels are cross-linked polymeric networks made of hydrophilic polymers and have widespread applications. Year after year, their application capabilities are expanding. With their ability to absorb and bind water, they can effectively control both their viscous and elastic properties. However, the stability of the hydrogel network depends on many factors, including the method of preparation. The simplest hydrogel structures are those obtained by physical methods. They are formed by environmental factors (temperature, pressure, pH) or physicochemical interactions (e.g., hydrophobic interactions or supramolecular chemistry), and characterized by a low stability and strength. On the other hand, hydrogels obtained by chemical transformations, especially the modern chemical transformations of polymers, allow materials with much better potential applications to be obtained [108,109]. In order to improve the parameters of synthetic and natural hydrogels, hybrid hydrogels have been developed, which allow the scope of application of this type of materials to be extended [109]. New-generation hydrogels are used primarily in medicine, including tissue engineering, drug delivery, reagent adsorption, tissue regeneration and gene therapies [110,111]. Thermosensitive chitosan–carbon nanotube hybrid hydrogels were developed as injectable materials for anticancer drug delivery. This drug delivery system can significantly enhance the development of targeted therapy strategies and sustainable chemotherapy [112]. Kowalska and coworkers [113] proposed new possibilities of using graphene oxide. They developed a freezing—gelling—thawing method for the fabrication of bulk gradient chitosan-matrix hydrogels. Composite systems modified with poly(ethylene glycol)-grafted-graphene oxide showed promising properties that can be successfully applied as tissue engineering scaffolds. In turn, Ganguly et al. [114] developed alginate-derived nitrogen-doped CDs as a drug carrier and toughening agent for hydrogels by a microwave-assisted method. According to authors the delayed network rupturing and superstretchability could make this material a valuable option for soft biomaterials and soft robotics. Serafin et al. [115], to create electroconductive and printable 3-D scaffolds, developed hybrid printable biomaterials comprised of alginate and gelatin hydrogel systems filled with carbon nanofibers (CNFs). Such a solution represents a significant contribution to the development of the biomedical field, in particular hybrid hydrogels as electro-conductive biomaterials.

Due to various hydrogel manufacturing techniques, their application possibilities have expanded considerably. Ehtesabi et al. [111] provide an easy-to-use tetracycline detection method, by encapsulated CDs in sodium alginate hydrogel. The encapsulation of carbon dots in the hydrogel structure broadens their use as tetracycline sensors and adsorbents for environmental tetracycline pollutants. Recently, Wang et al. [116] synthesized electro-conductive hydrogels (ECHs) by combining polyvinyl alcohol-borax (PVA) hydrogel matrix and 2,2,6,6-tetramethylpiperidine-1-oxyl (TEMPO)-cellulose nanofibers (TOCNFs), carbon nanotubes (CNTs), and polyaniline (PANI). Such hydrogels products have great application prospects in portable energy storage devices. In turn, Nayak et al. [117] obtained a hybrid hydrogel from polyvinylpyrrolidone (PVP) cross-linked by carbon quantum dots (CD) for simultaneous dye adsorption (both cationic and anionic), photodegradation and bacterial elimination from wastewater. The effect of the obtained hydrogel was the generation of reactive oxygen species (ROS) by CD embedded in the hydrogel matrix after exposure to sunlight, and the subsequent degradation of the adsorbed dyes. Furthermore, CD-induced ROS effectively destroyed both Gram-positive and Gram-negative bacteria in contaminated water less than 10 min after the photoexcitation of the hydrogel. Another example of the use of hydrogels in water treatment processes is the environmentally friendly (polymer–carbon dot) hybrid composites proposed by Jlassi et al. [118] used for removal of Cd^2+^ from industrial wastewater. Novel photoluminescent CDs were produced from petroleum coke waste, rich in oxygen, nitrogen, and sulfur functional groups and introduced into the chitosan polymer matrix.

It is also worth noting the large group of hydrogels with germicidal or bacteriostatic properties. Recently, a green method has been employed for the preparation of reduced graphene oxide (rGO) in nanocomposite polyelectrolyte complex (NC-PEC) hydrogel made of xanthan gum (XG) and chitosan (CS). Obtained hydrogel resulted in higher antibacterial activity against Gram-positive and Gram-negative bacteria and have specific wound dressing applications [119].

There are many more possibilities of using hydrogels, but it is not possible to list all of them here. This sector is developing intensively, still surprising us with something extraordinary.

### 4.2. Foams

The carbon-based nanomaterials can be used as components in other, higher 3-D nano and macro structures, such as foams or emulsions. These structures gain new mechanical, viscous-elastic, yield stress, biomedical and functional properties due to the appropriate dispersion of a special solid particles. Furthermore, introducing these one- or two-dimensional particle nanomaterials to higher three-dimensional spatial structures allows for the improvement and modification of their initial properties. Physicochemical and mechanical properties of classical foams are mostly a consequence of the irreversible decomposition of these systems, related to the processes of liquid drainage, coalescence or gas diffusion. However, the addition of suitable particles may slow down foam destruction or stop it completely. The lifetime of such a system can also be accurately controlled with the help of special “smart” particles. In some foams, specially designed particles can be used, which initially stabilize them but later help us to destroy them. The stabilization of aqueous foams by nanoparticles is a relatively new scientific discipline. People are still gathering new data and finding general rules for describing the stability and rheology of “particle-laden foams”. Hence, the influence of particle size, shape, structure, particle composition (mineral lor organic), concentration, wettability, pH, surface charge, and many other factors on foamability and foam stability must be carefully studied and analyzed. These problems have been already described for several systems and various solid and colloidal particles in the review by Krzan et al.: “Foams stabilized by particles” [120].

Graphene can also be used in solid foams in combination with other carbon-based nanomaterials. A similar use is three-dimensional porous nitrogen-doped graphene aerogels, which were synthesized using graphene oxide and chitosan nanoparticles [121]. The developed materials possessed hierarchical pores with a wide size distribution ranging from mesopores to macropores. After carbonization at different temperatures, the obtained aerosols demonstrated excellent electrochemical performance. Therefore, the authors believe that they can work as a supercapacitor electrode. Graphene flat particles can also be used together with fullerene-like spheres to produce higher spatial nano and macrostructures with new properties. For example, Zhao at al. [122] described that composite foam material controlling the concentration, size, and shape of fullerene-like spheroids with matching topological connectivity with graphene layers leads to new yield and mechanical properties.

The introduction of fullerenes to biodegradable polyurethane (PU) foams allows a material with better mechanical and physicochemical properties to be obtained [123]. Such a material was presented recently by Thangavelu at al., who developed a simple one-shot formation process of PU/fullerene foams. Transmission electron microscopy data confirmed the uniform distribution of fullerenes in obtained nanocomposites, while tensile strength was found to increase gradually due to the addition of fullerenes into the polyurethane-fullerene nanocomposites. Recently, Zhang at al. [124] created a method for fabricating carbon nanotube (CNT) foam. This is a composite porous carbon structure with controllable cell shape and distribution and, therefore, tunable properties, including density, porosity, elasticity, conductivity, and strength. Moreover, new materials offer additional advantages, such as mechanical flexibility and robustness, electrical conductivity, thermal stability, and resistance to the harsh environment compared with conventional foams.

### 4.3. Films

Synthetic nanomaterials (e.g., graphene, carbon nanotubes, mineral nanoparticles, and metallic nanoparticles) can be effectively integrated with biopolymers to produce films with desired properties such as electrical and thermal conductivity, barrier properties, and unique optical properties [125]. Composites of chitosan and carbon nanotubes (CNTs) are of increasing concern due to their attractive structural, mechanical, and electrical properties, which can find applications in catalysis, tissue engineering, biomedical and sensor applications, as well as biosensor fabrication [126]. Ahmed A.Alshahrani and colleagues prepared nanofiltration membranes from pristine and functionalized multiwall carbon nanotubes/biopolymer composites for water treatment applications [127]. The samples were made from a biopolymer (chitosan) containing functionalized multiwall carbon nanotubes (MWCNTs) with –NH_2_ and −COOH moieties. The comprehensive characteristics of all biopolymer membranes were systematically investigated. The results indicate that the properties of the biopolymer membrane were significantly affected as the top layer by the type of multi-walled carbon nanotubes. With MWCNTs−COOH/chitosan membranes, the highest permeability towards water was achieved, while the MWCNTs-NH_2_/chitosan membranes provided the best performance in salt rejection by properly balancing amine groups on the top layer, which could be separate monovalent and multivalent cations from salt solutions. Yamakawa A. et al. [128] investigated the structure and physical properties of composite films of cellulose nanofibers and multi-walled carbon nanotubes (CNF-MWCNTs) prepared by aqueous dispersion of MWCNTs using 4-O-methyl-α-d-glucuronoxylate as a MWCNT dispersion aid. The composite film exhibited a high electrical conductivity, good mechanical properties and a low thermal expansion coefficient. FE-SEM (field emission scanning electron microscopes) imaging showed that the carbon nanotubes dispersed homogeneously and formed a network to reinforce the cellulose nanofiber matrix. The improvement in the physical properties of the cellulose nanofiber film by the addition of MWNTs was caused by the composite structure. The obtained CNF-MWCNT hybrid film could be used for electromagnetic shielding or as a static free material. The article published by Jamróz and colleagues [129] presents the preparation of nanocomposite films from furcellaran (FUR) and nanofillers: graphene oxide (GO), carbon nanotubes (MWCNT) and silver nanoparticles. The presence of nano-additives in films improved their mechanical and thermal properties.

Numerous studies were conducted to improve the mechanical properties of polysaccharide-based films. One possible approach is to enrich the polysaccharide matrices with graphene oxide. One of the difficulties in the synthesis of graphene-based materials, which has limited their use in many fields, is the tendency to aggregation resulting from the presence of Van der Waals interactions between graphene layers. The introduction of metal nanoparticles on their surface prevents aggravation and also provides the good photoconductivity and catalytic properties of graphene/metal nanocomposites. Recently [10], films containing graphene and nanosilver in the hyaluronic acid matrix, and graphene, nanosilver and nanogold in the sodium alginate matrix, were synthesized. The films containing nanosilver/graphene oxide in hyaluronic acid matrix exhibited bacteriostatic activity against *E. coli*, Staphylococcus spp. and Bacillus spp. and showed a cytotoxic effect against WM266-4 human melanoma cell lines [10]. The incorporation of graphene oxide sheets within the sodium alginate matrix has a beneficial effect on the thermal stability of nanocomposite films [65]. The obtained films underwent the hydrolysis reaction faster and hydrolyzed to a greater extent than the alginate itself. Prepared composites containing silver nanocomposites exhibited slightly greater hydrophobic properties, a lower dispersion and surface free energy, and bacteriostatic activity against tested microorganisms [65]. Another work describes the synthesis of bionanocomposites, enriched with GO nanoparticles and based on a starch/chitosan binary matrix [11]. DSC analysis confirmed the influence of GO on the thermal properties of nanocomposites and higher melting points determined in samples containing GO. Mechanical tests showed that films are more resistant to cracking and showed high flexibility/extensibility. The introduction of GO significantly improved the elongation of the nanocomposite at break. The GO-containing nanocomposites showed bacteriostatic activity, but importantly, no toxicity to human cells was confirmed, which is a great advantage and extends the range of possible applications in various industries. The largest graphene addition used in the experiment was 3.0% of the total film-forming solution. The cell viability assays demonstrated that incubation with graphene nanocomposites was well-tolerated by human skin keratinocytes and human liver-derived cell line HepG2. This lack of cytotoxicity is very important, because numerous previous studies reported a considerable level of cytotoxicity in various cell lines [11].

In another work, Keleshteri et al. [130] designed a system that penetrates bone morphogenetic protein cells with the ability to image and deliver the protein. To optimize the scaffold ability in terms of physicochemical properties and cellular behavior, pectin microparticles combined with protein–carbon dots nanomaterial hybrids were incorporated into freeze-dried gelatin-elastin-hyaluronic acid-based composite scaffolds. The biological responses of the effects of the fabricated composite scaffolds on human osteosarcoma cell lines were investigated in terms of biocompatibility, cellular uptake, cell attachment, alkaline phosphatase activity and matrix mineralization production. The obtained results indicated the potential application of the obtained protein–CQDs polymeric materials for therapeutic and bone tissue engineering applications. By combining sodium alginate with CQDs, fibers and membranes with fluorescent properties were obtained [131]. Depending on the wavelength of the excitation radiation, the membrane emitted bright light of different colors, which will enable the resulting materials to be used in bioimaging, medical diagnostics, and in many industries for product labeling (preventing product adulteration). In addition, sodium alginate-CQDs nanofibers showed good biocompatibility and non-toxicity, suggesting the possibility of using nanofibers as wound dressing materials.

Other possible applications of polysaccharides as carbon nanoparticle carriers are summarised and presented in Table 1.

## 5. Future Prospective

Due to their unique properties, such as a high Young’s modulus, light weight, high thermal and electrical conductivity, good diffraction strength, mobility of charge carriers, large planar surface, electron delocalization, and biocompatibility, carbon nanomaterials in combination with biopolymers and particularly polysaccharides (which are generally available, biocompatible, biodegradable, renewable, inexpensive and non-toxic) can improve the quality and safety of food, and be used in medical and biological sciences, cosmetology and many other areas of our lives. Designing and investigating the properties of new materials presents researchers with a number of opportunities but also many challenges. These achievements will have a significant impact on the quality and safety of people’s lives by reducing environmental pollution, soil, water and air pollution, as well as food waste. 

## Figures and Tables

**Figure 1 polymers-14-00948-f001:**
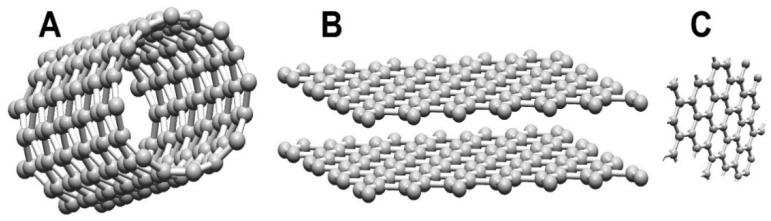
Carbon structures: (**A**) nanotubes, (**B**) graphene and (**C**) quantum dots.

**Table 1 polymers-14-00948-t001:** Selected applications of polysaccharides as carbon-based nanoparticles carriers.

Carbon Nanoparticles	Polysaccharides	Application	Ref.
Graphene and its derivatives	Chitosan/xyloglucan	Nanocarrier for biomedical applications	[132]
	Chitosan/carboxymethylcellulose		[133]
	Starch	Immunoassay to detect neuron-specific enolase with a triple signal amplification strategy	[134]
	Chitosan	Additives for water-based lubrication	[135]
	Dialdehyde cellulose	Film against COVID-19	[136]
	Chitosan	Microextraction of organic pollutants	[137]
	Chitosan/sodium alginate	Wastewater treatment	[138]
	Starch	Supercapacitor electrodes and efficient adsorbents	[139]
	Dextran	High-performance sodium batteries	[140]
Carbon nanotubes	Gelatin	Multifunctional robotic skin	[141]
	Konjac glucomannan	Scaffolds for muscle and cardiac nerve tissue regeneration	[142]
	Chitosan	Tissue engineering and biomedical applications	[143]
		Electrolyte membranes	[144]
	Cellulose	Smart papers for multifunctional sensing	[145]
		Scaffold for bone regeneration	[146]
		Thermal energy storage	[147]
	Alginate	Membranes for water filtration	[148]
			[149]
Carbon quantum dots	Starch/chitosan	Elements of smart and active packaging	[102]
	Pectin/gelatin		[150]
	Gelatin/carrageenan		[151]
	Sodium alginate	Tetracycline fluorescent sensor and adsorber	[111]
	Starch	Foil for monitoring spoilage of pork	[152]
	Gelatin	Sensors for label-free breast cancer detection	[153]
	Chitosan	Wound healing and drug delivery system	[73,107]

## Data Availability

Not applicable.

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
