# Peer review of "Polysaccharides Composite Materials as Carbon Nanoparticles Carrier"

_polymers, 2022, doi:10.3390/polym14050948_

Round 1
Reviewer 1 Report
The manuscript “Polysaccharides composite materials as carbon nanoparticles carrier”, intended to presents the progress of designing innovative carbon-base nanomaterials and applications of nanotechnology in diverse fields. The comments and problems are as follows:
- There is no mention of polysaccharides in the introduction.
- Some mistakes should be revised, on line 135 and 137.
- Weather the packaging and nanosensors are carbon nanoparticles (line 256-262)?
- The author compared some electrode materials (line 281-365), but what is the relationship between polysaccharides and carbon-based materials in energy storage?
- The GO-containing nanocomposites showed bacteriostatic activity, but, no toxicity to human cells was confirmed (on line 560-562), I wonder what is the dose of this material?
Author Response
Dear Reviewer,
Thank you very much for your constructive comments. We have revised and corrected our manuscript according to your suggestions. All the modifications are highlighted in red. Below we present the itemized list of the introduced changes.
The manuscript “Polysaccharides composite materials as carbon nanoparticles carrier”, intended to presents the progress of designing innovative carbon-base nanomaterials and applications of nanotechnology in diverse fields. The comments and problems are as follows:
There is no mention of polysaccharides in the introduction.
The introduction section has been revised according to the Reviewer’s suggestions.
Some mistakes should be revised, on line 135 and 137.
It was corrected according to Reviewer suggestions.
Weather the packaging and nanosensors are carbon nanoparticles (line 256-262)?
Thank you for this suggestion. We corrected the sentence.
The author compared some electrode materials (line 281-365), but what is the relationship between polysaccharides and carbon-based materials in energy storage?
The explanation was added according to Reviewer suggestions.
The GO-containing nanocomposites showed bacteriostatic activity, but, no toxicity to human cells was confirmed (on line 560-562), I wonder what is the dose of this material?
The largest graphene addition used in experiment was 3.0% of the total film-forming solution. The cell viability assays demonstrated that incubation with graphene nanocomposites is well tolerated by human skin keratinocytes and human liver-derived cell line HepG2. This lack of cytotoxicity is very important, because numerous previous studies reported a considerable level of cytotoxicity in various cell lines.
Best regards,
Authors
Reviewer 2 Report
Dear all,
Greetings
Please find enclosed my comments regarding paper
Referenced as: polymers-1599180
Titled: Polysaccharides composite materials as carbon nanoparticles carrier
The authors have performed good and impressive review concerning the polysaccharides composite and using these composites in many applications in nanotechnologies, but this review can be accepted for publication in Polymers, after adressing and answering all these comments (Minor Revisions)
1) Title: ok
2) Abstract: please add the best conditions for your composite, graphene, nanocarbones or others
3) Keywords: add more keywords
4) Comments:
- you discuss in the introduction about nanotechnology please add the appropriate references s American physicist Richard Feynman [X], [1-3], [6-8], [9-12], [12-15]……
- 2.1. Graphene
- Table 1 splited between two pages
- For each materials add figures showing the form
- References ok
With regards
Author Response
Dear Reviewer,
Thank you very much for your constructive comments. We have revised and corrected our manuscript according to your suggestions. All the modifications are highlighted in red. Below we present the itemized list of the introduced changes.
Dear all,
Greetings
Please find enclosed my comments regarding paper
Referenced as: polymers-1599180
Titled: Polysaccharides composite materials as carbon nanoparticles carrier
The authors have performed good and impressive review concerning the polysaccharides composite and using these composites in many applications in nanotechnologies, but this review can be accepted for publication in Polymers, after adressing and answering all these comments (Minor Revisions)
1) Title: ok
Thank you.
2) Abstract: please add the best conditions for your composite, graphene, nanocarbones or others
Carbon nanoparticles have a wide range of applications, they can react or be functionalized with various biopolymers, therefore the conditions for obtaining composites, their amount, as well as their effect will depend on the above-mentioned parameters. The best conditions are chosen when designing specific materials. In the introduction section we added additional information concerning this matter (please see the text in red). We hope that we have given a satisfying answer on this point.
3) Keywords: add more keywords
It was corrected according to Reviewer suggestions.
4) Comments:
- you discuss in the introduction about nanotechnology please add the appropriate references s American physicist Richard Feynman [X], [1-3], [6-8], [9-12], [12-15]……
It was corrected according to Reviewer suggestions.
- 2.1. Graphene
It was corrected according to Reviewer suggestions.
- Table 1 splited between two pages
We decided that the table 1 should be changed into more appropriate to the title of the manuscript.
- For each materials add figures showing the form
It was corrected according to Reviewer suggestions.
- References ok
With regards
Thank you for this remark, we did our best to correct these issues.
Best regards,
Authors

Round 2
Reviewer 1 Report
The revised manuscript has been revised. I think that the revised manuscript can be accepted in the present form.